# *Siraitia grosvenorii* (Swingle) C. Jeffrey: Research Progress of Its Active Components, Pharmacological Effects, and Extraction Methods

**DOI:** 10.3390/foods12071373

**Published:** 2023-03-23

**Authors:** Jiajing Duan, Dong Zhu, Xiuxia Zheng, Yang Ju, Fengzhong Wang, Yufeng Sun, Bei Fan

**Affiliations:** 1Institute of Food Science and Technology, Chinese Academy of Agricultural Sciences, No. 2 Yuan Ming Yuan West Road, Beijing 100193, China; 2Key Laboratory of Agro-Products Quality and Safety Control in Storage and Transport Process, Ministry of Agriculture and Rural Affairs, Beijing 100193, China

**Keywords:** bioactive component, extraction, pharmacological effect, *Siraitia grosvenorii*

## Abstract

*Siraitia grosvenorii* (Swingle) C. Jeffrey, a perennial vine of the Cucurbitaceae family, is a unique medicine food homology species from China. *S. grosvenorii* can be used as a natural sweetener in the food industry and as a traditional medicine for moistening the lungs, quenching a cough, smoothing the intestines, and relieving constipation. Additionally, the fruits, roots, stems, and leaves of *S. grosvenorii* are rich in active ingredients, and have pharmacological effects such as immune regulation, hypoglycemia, and antioxidant, hepatoprotective, and antitumor effects, etc. Therefore, *S. grosvenorii* has broad application prospects in the pharmaceutical industry. This paper reviews the bioactive components, pharmacological effects, and extraction methods of *S. grosvenorii*, summarizes them, and proposes their future development directions. This current overview highlights the value of *S. grosvenorii*. By documenting the comprehensive information of *S. grosvenorii*, the review aims to provide the appropriate guidelines for its future in-depth development and the utilization of *S. grosvenorii* resources for their roles as active ingredient (triterpenoids, flavonoids, and polysaccharides, etc.) sources in the food industry and in the development of functional foods.

## 1. Introduction

*Siraitia grosvenorii* (Swingle) C. Jeffrey (Figure 1A) is a perennial vine of the Cucurbitaceae family [1]. *S. grosvenorii* has relatively high environmental requirements for its growth. Its main production areas are Lingui and Yongfu in China’s Guangxi province, which produce more than 90% of the world’s *S. grosvenorii* output [1]. The ripe fresh fruits of *S. grosvenorii* are green (Figure 1B). Their skin and flesh become brown after baking, saccharification, and drying (Figure 1C).

*S. grosvenorii* was considered to be one of China’s first medicine food homology species in 2002. It is an essential source of natural sweeteners, and its unique composition, mogroside, has the characteristics of pure nature, excellent taste, zero calories, and high sweetness, with a sweetness of about 300 times that of sucrose, and a taste that is close to sucrose [2]. In 1995, the FDA approved the addition of mogroside to foods, and in 1996, China approved its use as an additive in food products. *S. grosvenorii* has promising applications in the food and pharmaceutical industries. Research on traditional medical has revealed that *S. grosvenorii* could moisten the lungs, quench cough [3], relax the intestine, and relieve constipation [4]. According to previous studies, more than 100 compounds have been isolated from *S. grosvenorii*, including at least 46 kinds of triterpenoids, 7 kinds of flavonoids, 19 kinds of amino acids, and 2 types of polysaccharides [5]. Modern pharmacological studies have shown that the active ingredients in *S. grosvenorii* could regulate immunity [6], lower the blood sugar [7], antioxidant activity [8], and antitumour agents [9], protect the liver, and prevent tooth decay [10], etc. (Figure 2). The active ingredients in *S. grosvenorii*, such as cucurbitane triterpenoids, flavonoids, and polysaccharides, have been intently investigated. The active ingredients and pharmacological effects of *S. grosvenorii* are shown in Figure 3. With the enhancement of public health awareness, there is an increasing demand for health products that have been produced with *S. grosvenorii* as their raw material.

This paper focuses on a detailed review of the bioactive components, pharmacological effects, and extraction methods of *S. grosvenorii*, and proposes a future development direction. The review aims to provide appropriate guidelines for the in-depth development and utilization of *S. grosvenorii* resources in the future.

## 2. Triterpenoids

### 2.1. Triterpenoid Ingredients in S. grosvenorii

Triterpenoids are the main bioactive components of *S. grosvenorii*. They are sweet ingredients of *S. grosvenorii*, with a content of 3.76–3.86% in the dried fruit [11]. Currently, 12 triterpenoids (cucurbitane glycosides) have been found in *S. grosvenorii*, namely: siamenside I, grosmomoside I, mogroside IIE, mogroside III, and mogroside IIIE, mogroside IV, mogroside V, 11-oxo-mogroside V, mogroside VI, mogroester, mogroside A, and mogroside neoside [12,13]. Their molecular structures are shown in Figure 4. Peng et al. established an ultra-performance liquid chromatography method (UPLC) for determining these different triterpenoid ingredients in *S. grosvenorii*, showing that the mogroside V content is the highest. Mogroside V accounts for about one fifth of the total glycosides, which are 425 times sweeter than glucose [14,15]. Therefore, mogroside V is often used as a natural sweetener in the food industry.

### 2.2. Pharmacological Effects of Triterpenoids in S. grosvenorii

Modern pharmacological studies have shown that mogrosides can regulate immunity, lower the blood sugar and lipid levels, protect the liver, and are antioxidants and antitumour agents. Mogrosides have various beneficial effects for both type 1 and type 2 diabetes. Chen et al. found that mogrosides modulate the antigen expression of the splenic lymphocytes in type 1 diabetic mice, which, in turn, had a therapeutic effect on type 1 diabetes [16]. Liu et al. used a high-fat diet, combined with a streptozotocin (STZ)-induced diabetes model, to investigate the hypoglycemic and hypolipidemic effects of glycoside-rich extracts (MGE) and their potential mechanisms. Their results showed that MGE benefits type 2 diabetic mice, including by lowering their FBG and increasing the serum insulin, inhibiting gluconeogenesis and insulin resistance, enhancing hepatic fatty acid oxidation, improving blood lipids, and impairing hepatic AMPK and insulin signalling. The hypoglycemic and hypolipidemic activities of MGE may be attributed to the attenuation of insulin resistance and the activation of hepatic AMPK signalling [17]. Mogroside V can effectively regulate the insulin resistance of HepG2 cells and improve the blood glucose levels, insulin sensitivity, glucose and lipid metabolism disorders, and oxidative stress in T2DM rats. It can also effectively alleviate insulin resistance by regulating the PI3K/Akt pathway in the liver and fat of these T2DM rats, regulating their Akt phosphorylation, promoting glycogen synthesis, and enhancing the glucose transport capacity [18]. Chen et al. investigated the metastatic efficiency of mogroside V in lung cancer cells that were cultured in a hyperglycemic state. They found that mogroside V inhibited the invasion and migration that is induced by hyperglycemia through upregulating the E-Cadherin expression and downregulating the N-Cadherin, Vimentin, and Snail expression. Additionally, under hyperglycemic conditions, mogroside V can break microfilaments and reduce the expressions of Rho A, Rac1, Cdc42, and p-PAK [19]. Liu et al. investigated the effect of mogroside IVe on the proliferation of colorectal cancer HT29 cells and laryngeal cancer Hep-2 cells. Their results showed that mogroside IVe induced apoptosis by upregulating p53 and downregulating p-ERK1 and MMP-9, inhibiting the proliferation of the colorectal and laryngeal cancer cells [20]. They also demonstrated that mogrosides could inhibit tumour growth by disturbing the growth cycle of pancreatic cancer cells and inducing cell death [21]. Numerous studies have reported that mogrosides possess anti-inflammatory activity. Di et al.’s study used a mouse ear swelling model and reported that mogrosides exerted anti-inflammatory effects by inhibiting the expression of key inflammatory genes, promoting the expression of anti-inflammatory genes, and suppressing cellular inflammatory responses [22]. Mogrosides could improve non-alcoholic steatohepatitis by preventing liver fat accumulation and inhibiting fat peroxidation [23]. Song et al. studied the wheezing effect of mogroside V on ovalbumin (OVA)-induced asthmatic mice, and found that mogroside V administration was effective in attenuating OVA-induced airway hyperresponsiveness and reducing the number of inflammatory cells in bronchoalveolar lavage fluid (BALF) [24]. Mogroside IIIE could exert protective effects against lipopolysaccharide-induced acute lung injury, which shows its therapeutic potential in clinically treating acute lung injury [25]. Mogroside V could significantly reduce the ammonia stimulation in mice respiratory systems, decrease cough, and increase the red phenol excretion in the trachea, indicating that mogroside V possesses a cough expectorant effect [26]. In addition, Ju et al. showed that mogroside V could effectively improve the schizophrenic behaviour of mice with a low glutamate function and modulate partial permanent impairment [27].

### 2.3. Extraction of Triterpenoids from S. grosvenorii

The initial extraction method of mogroside was a direct boiling extraction or ethanol extraction. The hot water extraction could obtain the mogrosides with a high yield. This method is mature, stable, simple, low cost, and the product quality is better than that from the ethanol or other solvent extractions [28]. Chen et al. used water as a solvent and determined the optimal extraction process of the mogrosides as follows: a material-liquid ratio of 1:15 (g/mL), soaking for 30 min, and then extracting three times for 60 min each. The yield of the mogrosides reached 5.6%. This process is stable, feasible, and suitable for industrial production [29]. Yan et al. used ethanol as a solvent, optimising the process based on the results of five single-factor tests: the ethanol volume fraction, extraction temperature, extraction time, number of extractions, and liquid-to-material ratio. They conducted response surface analysis experiments on three factors: the ethanol volume fraction, extraction temperature, and liquid-to-material ratio, with the extraction rate of mogroside as the index of investigation. The optimal extraction process of the mogrosides was as follows: 50% ethanol as the solvent, a material-liquid ratio of 1:20 (g/mL), an extraction temperature of 60 °C, shaking for 100 min, and then extracting three times. The resultant yield of the mogrosides was 5.9% [30]. The ultrasonic-assisted extraction method uses linear alternating vibrations of ultrasound and radiation pressure along the direction of the sound wave propagation, in order to disrupt the cell and cell membrane structure of the raw material of *S. grosvenorii*, and to promote a faster and more efficient extraction of the mogrosides [31]. Song used this ultrasonic-assisted extraction method to extract the mogrosides and determined the optimal extraction conditions (as follows) with single-factor and orthogonal tests: an ethanol volume fraction of 60%, a material-liquid ratio of 1:45 (g/mL), an ultrasonic temperature of 55 °C, an ultrasound frequency of 40 kHz, and an ultrasonic time of 45 min. The yield reached 2.98% [32]. Zhu et al. established a microwave extraction method for the mogrosides and found that it was significantly better than the conventional boiling method [33]. The flash extraction method is a new technology used for natural active compound preparation. The principle involves using high-speed mechanical shear force and super-speed dynamic molecular penetration in the presence of an appropriate solvent, in order to rapidly destroy the cell tissue so that the chemical components (or active ingredients) inside the tissue cells can quickly reach the balance, inside and outside of the tissue, which are then are extracted through filtration. This flash extraction method is highly efficient, rapid, requires no heating, is not destructive, and is suitable for various solvents, compared with traditional methods [34]. Liu et al. determined the flash extraction process of the mogrosides with orthogonal experiments and used high-performance liquid phase (HPLC) detection methods to confirm the purity. The optimum extraction conditions were obtained and are as follows: a material-liquid ratio of 1:20 (g/mL), a blade speed of 6000 r/min, a temperature of 40 °C, and an extraction time of 7 min. The yield of the mogrosides was 6.9%, with a purity above 92% [35]. The extraction methods of the mogrosides from *S. grosvenorii* are summarised in Table 1. From Table 1, it can be seen that the extraction rate and purity of the flash extraction are better than the other extraction methods.

## 3. Flavonoids

### 3.1. Flavonoid Compounds in S. grosvenorii

Flavonoids belong to the natural polyphenolic compound class that largely exists in plants. There are relatively few reports about the flavonoids in *S. grosvenorii*, and most of them were mentioned in the analysis of the mogrosides. Chen et al. determined that the content of the total flavonoids in each *S. grosvenorii* was 5–10 mg [36]. Kaempferol and quercetin are the main flavonoids in *S. grosvenorii*. Si et al. isolated two flavonoid glycoside components, kaempferol-3, 7-a-L-dirhmnopyranoside and grosvenorine, from a fresh *S. grosvenorii* aqueous extract [37]. Huang et al. isolated five flavonoids from the *S. grosvenorii* flower, including kaempferol, kaempferol-7-O-L-rhamnoside, kaempferol-3-O-L-rhamnoside 7-O-[β-D-glucosyl-(1-2)-α-L-rhamnoside], 3-O-L-rhamnoside, and 3-O-D-glucopyranoside, and two flavonoids from the *S. grosvenorii* leaf, including kaempferol 3,7-di-O-L-rhamnoside (kaempferol) and quercetin 3-O-D-glucopyranoside 7-O-L-rhamnoside [38,39,40] (Figure 5).

### 3.2. Pharmacological Effects of Flavonoids in S. grosvenorii

The flavonoids in *S. grosvenorii* show vigorous antioxidant activity due to their many phenolic hydroxyl groups, which achieve the antioxidant capacity by scavenging reactive oxygen species and free radicals. Huang et al. found that the *S. grosvenorii* flavonoids exhibited significant 2,2-diphenyl-1-picrylhydrazyl (DPPH) scavenging activity [41]. The flavonoids in the *S. grosvenorii* leaf showed vigorous antioxidant activity, four times more potent than that of butylhydroxytoluene (BHT) [42]. The flavonoids in *S. grosvenorii* also show significant antimicrobial activity. Zhang et al. confirmed that the flavonoids in *S. grosvenorii* exhibited a substantial inhibitory effect on Staphylococcus aureus, Bacillus subtilis, Clostridium perfringens, Pseudomonas aeruginosa, Escherichia coli, Candida albicans, and Aspergillus niger [43]. The flavonoids in *S. grosvenorii* could effectively reduce the incidence of diabetes. Li et al. found that the flavonoids in *S. grosvenorii* significantly lowered the blood glucose levels, inhibited α-glucosidase, and protected the pancreas [44]. In addition, the flavonoids in *S. grosvenorii* could effectively alleviate sports fatigue. Chen et al. studied the effects of the flavonoids in *S. grosvenorii* on FLK-1, BFGF mRNA, and protein expression in the gastrocnemius muscle of trained rats. They demonstrated that a supplementation with flavonoids during exercise could increase the blood supply capacity and improve the oxygen transport capacity of the muscle tissues [45].

### 3.3. Extraction of Flavonoids from S. grosvenorii

The main extraction methods of the flavonoids from *S. grosvenorii* are solvent reflux and microwave and ultrasonic extractions. Zhang et al. used solvent reflux extraction to optimise the extraction process of the flavonoids from *S. grosvenorii.* He used the ethanol volume fraction, extraction temperature, extraction time, and material-liquid ratio as the variables, and the flavonoid extraction amounts as the response values. The optimum solvent reflux extraction conditions were obtained: an ethanol volume fraction of 88%, an extraction temperature of 80 °C, an extraction time of 118 min, and a material-liquid ratio of 1:27 (g/mL). The yield of the flavonoids was 0.56% [43]. Qin et al. determined the optimum microwave-assisted process conditions for the extraction of the flavonoids from *S. grosvenorii*: an ethanol volume fraction of 50%, a material-liquid ratio of 1:35 (g/mL), a microwave power of 650 W, and an extraction time of 25 min. The yield of the flavonoids was 1.72%, which was higher than that of the solvent reflux extraction method [46]. Li et al. determined the optimum ultrasonic process conditions for the extraction of the flavonoids from *S. grosvenorii* via a response surface method: an ultrasound temperature of 50.65 °C, an ultrasound time of 29.2 min, and a material-to-liquid ratio of 1:34.75 (g/mL). The yield of the flavonoids was 2.25%. This method is simple, time-saving, and efficient [47]. The extraction methods of the flavonoids from *S. grosvenorii* are summarised in Table 2 [48,49,50,51,52,53]. For the *S. grosvenorii* flavonoids, the yield of the ultrasonic extraction was significantly higher than that of the other extraction methods, and the optimal extraction process conditions were: an ethanol volume fraction of 80%, a liquid-to-material ratio of 38:1, and an extraction time of 104 min.

## 4. Polysaccharides

### 4.1. Polysaccharides in S. grosvenorii

Polysaccharides, essential carbohydrates in nature, are polymeric carbohydrate macromolecules consisting of long-chain monosaccharide units that are linked by glycosidic bonds [54]. Sometimes, these macromolecules consist of hundreds or even thousands of monosaccharides, having molecular weights that range from several thousands to tens of thousands, or even tens of millions [55]. Polysaccharides largely exist in plants, animals, and microbes, and exhibit a variety of physiological functions. Most *S. grosvenorii* polysaccharides exist in the plant flesh [56]. In recent years, research on the structure of the polysaccharides in *S. grosvenorii* has gradually begun. Huang et al. isolated a polysaccharide (SGPS2, 0.65 × 103 kDa) from *S. grosvenorii*. SGPS2 consists of L-isomers of (1→2,4) linked rhamnose and (1→4) linked rhamnose residues in the main chain, and (1→2) linked rhamnose and (1→3) linked rhamnose in the side chains. Terminal residues attached the main chain to the rhamnose [57]. Zhu et al. characterised a new polysaccharide (SGP, 1.93 × 103 kDa) from an *S. grosvenorii* pomace. SGP consists of α-L-Arabinose, α-D-Mannose, α-d-Glucose, α-D-Galactose, glucuronic acid, and galacturonic acid, with a ratio of 1:1.92:3.98:7.63:1.85:7.34. The backbone of SGP was composed of galactoses and linked by a α-(1,4)-glycosidic bond. The branch chains consist of a α-1,6 linked glucose branch, α-1,6 linked mannose branch, α-1,3 linked galactose branch, and arabinose branched (α-L-Ara (1→) [58]. Gong et al. isolated a polysaccharide (SGP–1–1, 19.037 kDa) from *S. grosvenorii* that consists of galactose, mannose, and glucose, with a ratio of 1: 2.56: 4.90. It contains α- and β-glycosidic bonds and is elucidated as a glucomannan with a backbone composed of 4) -β-D-Glcp- (1→4) -, α-D-Glcp- (1→4) -, and 4) -Manp- (1 residue. α-1,6 linked an α-D-Galp branch, and α-1,6 linked an α-D-Glcp branch) [59]. The polysaccharide structure from *S. grosvenorii* is shown in Figure 6 [60]. Recently, the adoption of suitable methods to change the structure, molecular weight, and substituents of polysaccharides, in order to alter their physicochemical properties, enhance their bioactivities, or even obtain new bioactive derivatives, has become a hot research topic [61]. At present, the main modification approaches for polysaccharides are chemical (such as sulphation and carboxymethylation) [62], physical (such as ultrasonic degradation modifications and ion radiation modifications) [61], and biological (for example, enzymatic and genetic engineering) [63] methods. Bai et al. modified the *S. grosvenorii* polysaccharides via carboxymethylation and found that the derivative exhibited a more substantial antioxidant capacity than before the modification [64].

### 4.2. Pharmacological Effects of Polysaccharides in S. grosvenorii

Polysaccharides have attracted much attention due to their unique physicochemical properties. The polysaccharides in *S. grosvenorii* have antioxidant activity in vitro, especially in scavenging DPPH radicals. They can also decrease the reactive oxygen species (ROS) and the percentage of apoptotic and necrotic cells, in a dose-dependent manner in H2O2 oxide injury PC12 cells [58]. Zhang et al. found that the polysaccharides in *S. grosvenorii* could promote the proliferation of spleen cells and regulate the ROS levels in vitro. Furthermore, they significantly raised the spleen and thymus indices and superoxidase dismutase activity, and regulated the spleen and thymus cytokine levels in mice in vivo. The data suggested that the polysaccharides in *S. grosvenorii* possess immunomodulatory and antioxidant effects [65]. Zhang et al. showed that the polysaccharides in *S. grosvenorii* significantly enhanced the immune function of cyclophosphamide-induced immunosuppressed mice [66]. The administration of the polysaccharides in *S. grosvenorii* can significantly decrease the total plasma cholesterol, triglyceride, and glucose levels. Furthermore, they can restore the blood lipid levels of diabetic rabbits (*p* < 0.05). These results demonstrate that the polysaccharides in *S. grosvenorii* not only ameliorated the lipid disorder, but also lowered the plasma glucose levels [67]. Cui found that the *S. grosvenorii* polysaccharide, SGP–1–1, down-regulated the expression of TLR4, NF-κB p65 transcript and protein levels, and reduced oxidative stress and the inflammatory response in the body, exerting a protective effect on DN mice [68].

### 4.3. Extraction of Polysaccharides from S. grosvenorii

Hot water extraction is the most common approach for the preparation of water-soluble polysaccharides because it is simple and low-cost. Hot water extraction usually works together with alcohol precipitation, which can avoid the poor solubility of the polysaccharides in organic solvents. Cui applied a hot water extraction with ethanol precipitation to optimise the extraction process for the crude polysaccharides from *S. grosvenorii* by a single factor combined with response surface optimisation. Under a material-to-liquid ratio of 1:35 (g/mL), an extraction temperature of 70 °C, and an extraction time of 2.5 h, the yield was 6.56% [68]. The ultrasonic-assisted polysaccharide extraction uses the tremendous pressure that is generated by ultrasonic cavitation to cause the cell wall to rupture. In contrast, the vibration effect that is generated by ultrasonic waves accelerates the release and dissolution of the polysaccharides, which can markedly improve the polysaccharide yield. Chen et al. optimised this ultrasonic-assisted extraction process of the *S. grosvenorii* polysaccharides by a response surface method and determined the optimal conditions: a material-to-liquid ratio of 1:30 (g/mL), an ultrasonic temperature of 70 °C, an ultrasonic power of 300 W, and an ultrasonic time of 30 min. The yield of the *S. grosvenorii* polysaccharides was 15% higher than that of the hot water extraction method [69]. An enzymatic extraction could accelerate the dissolution of the polysaccharides after the enzymatic decomposition of the plant tissues, which is a mild condition. Li et al. determined the optimum conditions for extracting the *S. grosvenorii* polysaccharides with a cellulase method: an extraction temperature of 50 °C, an enzyme activity of 500 U/g, a hydrolysis pH of 6.0, and a hydrolysis time of 55 min. Under these optimal conditions, the yield of the *S. grosvenorii* polysaccharides was 10–22% higher than that of the traditional hot water extraction method [70]. Additionally, since a microwave-assisted extraction can significantly shorten the extraction time, continuous countercurrent extraction technology can extract some plant polysaccharides with a large viscosity, and membrane separation technology has the advantage of a high extraction rate and has been applied to *S. grosvenorii* polysaccharides extraction in recent years. The extraction methods of the polysaccharides from *S. grosvenorii* are summarised in Table 3 [71,72]. The extraction rate of the *S. grosvenorii* polysaccharides can reach up to 14.55%, which is when using the hot water extraction method. It is also a relatively simple method.

## 5. Proteins and Amino Acids

In the 1980s, Xu et al. investigated the types of nutrients that are in *S. grosvenorii* and found that the protein content is 7.11–10.78% in the dried fruit. Additionally, *S. grosvenorii* contains eighteen kinds of amino acids, and eight of them are essential amino acids for the human body. Amino acids play a vital role in the physiological regulation of the human body. It may be that peptides boost them to greater extent with regard to their nutraceutical and functional potential. Glutamic acid, an important material for synthesising glutathione and the non-essential amino acid with the highest human demand, exists in high levels in *S. grosvenorii* (108.2–113.3 mg/kg) [73].

## 6. Other Ingredients

The non-triterpene glycosides in *S. grosvenorii* are non-sugar sweetening substances. The D-mannitol in the fresh fruit of *S. grosvenorii* has a sweetness intensity that is equivalent to 0.55–0.65 times that of sucrose, and is used as a substitute for sweetened foods or flavouring agents for diabetic patients [74]. In addition, sixteen essential microelements and many macronutrients exist in the *S. grosvenorii* fruit and root. Its content of potassium, calcium, magnesium, and selenium is high, particularly in the fruit (12,290.8 ppm, 667.52 ppm, 549.96 ppm, and 0.1864 ppm, respectively) [75]. Each microelement and macronutrient have replaceable physiological functions and are essential for maintaining human metabolism [76].

## 7. Conclusions

With improved living standards, people’s attention to medicinal and food products has gradually increased. *S. grosvenorii*, as a leading food and medicinal product, serves as a daily dietary material and plays an important role in nutritional and health effects.

Due to current security concerns, consumers are demanding “natural” products that are free from organic solvents. Supercritical fluid extraction is environmentally friendly and safe. It has been successfully proposed for the extraction of active compounds from botanical matter and has proven to be effective in enhancing the extraction rate and quality of bioactive products. This is especially important for functional foods and the nutraceutical market [77,78]. Therefore, the application of supercritical fluid extraction to the extraction of the active ingredients from *S. grosvenorii* can be considered in the future.

The biological activities of *S. grosvenorii* are relatively widely reported, such as its apparent pharmacological effects as an antioxidant, hypoglycemic, cough suppressant, and antibacterial agent. However, the structural studies on the active ingredients of *S. grosvenorii* are still insufficient. Due to the diversity and complexity of its molecular structure, the relationship between its structure and biological activities has not been well established. Whether its active substances have synergistic effects with each other has not yet been investigated or reported. Therefore, the relationship between the structure and the bioactivity needs to be further investigated, and the mechanism of action of each active substance needs to be studied in depth to better understand its functional effects and outcomes. In addition, the pharmacological effects of *S. grosvenorii* are currently focused on cellular models and animal experiments, with fewer clinical studies. *S. grosvenorii* also contains several more complex chemical components, such as flavonoid glycosides, lignans, and fatty acid glycosides, which need further exploration.

In future studies, the conformational relationships of the active components in *S. grosvenorii* should be further explored. The multiple bioactive components in *S. grosvenorii* should be fully utilised to optimise its role as a novel drug raw material in the food industry and in functional food development.

## Figures and Tables

**Figure 1 foods-12-01373-f001:**
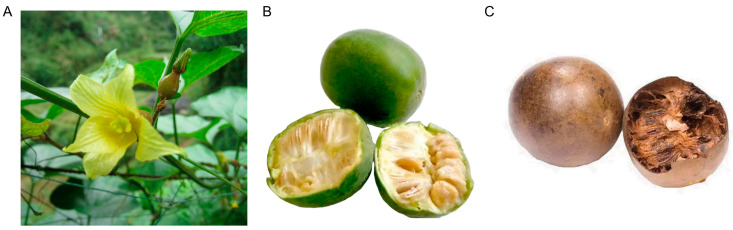
(**A**) The flowers, stems, and leaves of *S. grosvenorii*. (**B**) Fresh fruit of *S. grosvenorii*. (**C**) Dry fruit of *S. grosvenorii*.

**Figure 2 foods-12-01373-f002:**
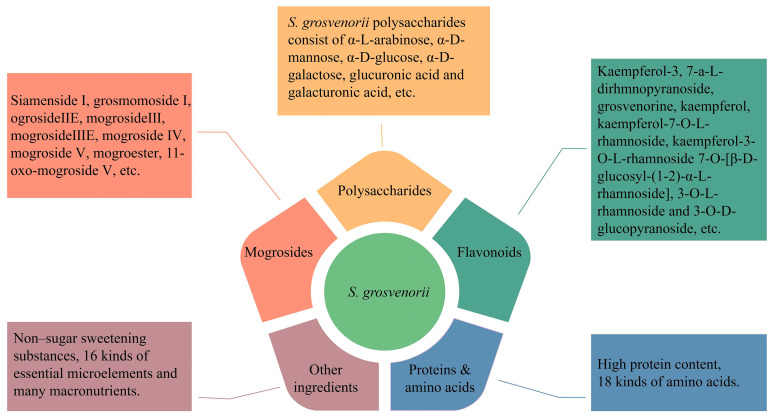
Representative bioactive compounds in *S. grosvenorii*.

**Figure 3 foods-12-01373-f003:**
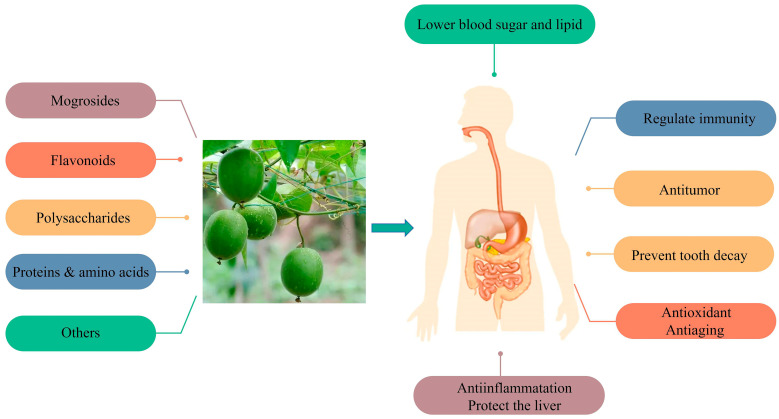
Pharmacological effects of *S. grosvenorii*.

**Figure 4 foods-12-01373-f004:**
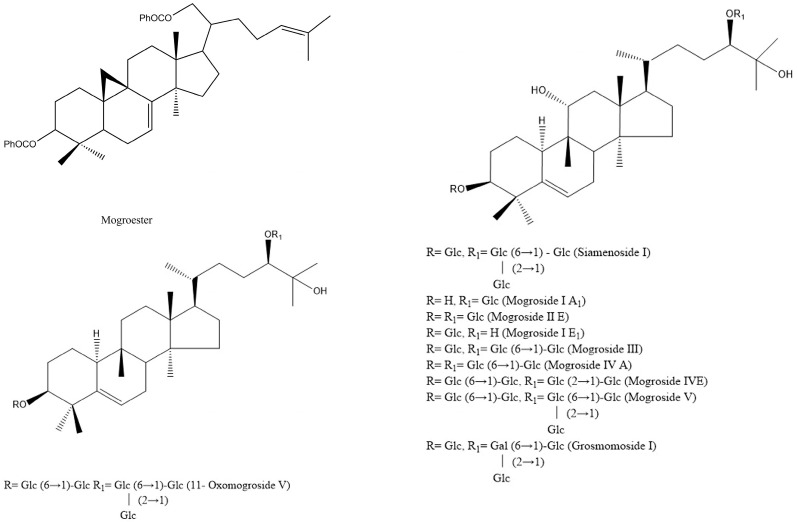
Structures of triterpenoids in *S. grosvenorii*.

**Figure 5 foods-12-01373-f005:**
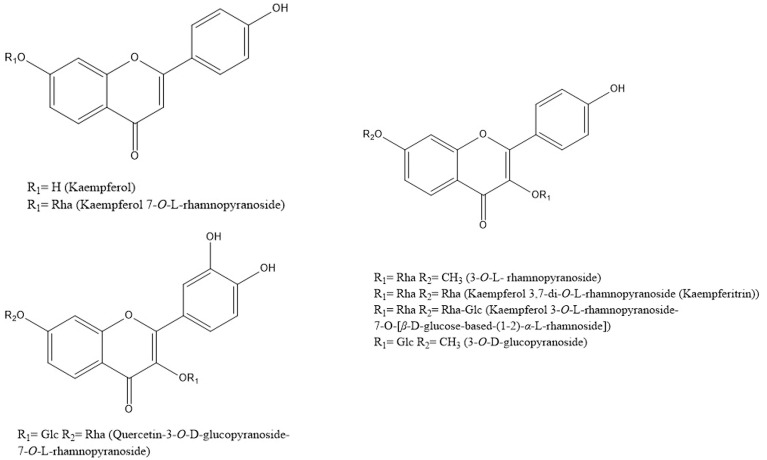
Structures of flavonoids in *S. grosvenorii*.

**Figure 6 foods-12-01373-f006:**
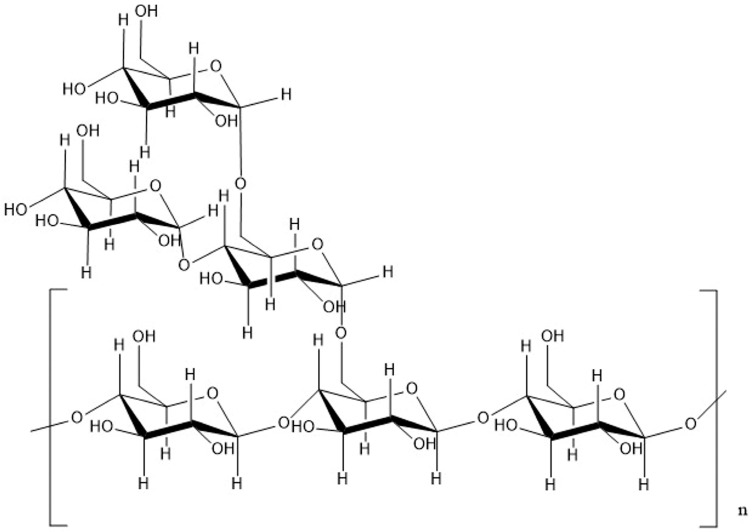
Structures of *S. grosvenorii* polysaccharide SGP–1–1.

**Table 1 foods-12-01373-t001:** Extraction methods of mogrosides from *S. grosvenorii*.

Method	Materials	Temperature (°C)	Time (min)	Material-Liquid Ratio (g/mL)	Other Conditions	Yield (%)	Ref.
Hot water extraction	Dry fruit		180	1:15	30 min per immersion	5.60	[29]
Ethanol extraction	Dry fruit	60	300	1:20	50% Ethanol	5.90	[30]
Ultrasonic extraction	Dry fruit	55	45	1:45	60% Ethanol Ultrasound frequency 40 kHz	2.98	[32]
Microwave extraction	Fresh fruit		15	1:8	300 MHz–300 GHz	4.80	[33]
Flash extraction	Dry fruit	40	28	1:20	Blade speed 6000 r/min	6.90	[35]

**Table 2 foods-12-01373-t002:** Extraction methods of flavonoids from *S. grosvenorii*.

Method	Materials	Temperature (°C)	Time (min)	Material-Liquid Ratio (g/mL)	Other Conditions	Yield (%)	Ref.
Reflux extraction	Dry fruit	70	120	1:15	70% Ethanol	0.58	[48]
	Dry fruit	80	60	1:30	80% Ethanol	1.52	[49]
	Dry fruit	80	118	1:27	88% Ethanol	0.56	[43]
	Dry leaves	Boiling	60	1:30	3 Times	1.32	[50]
Ultrasonic extraction	Dry fruit	70	40	1:25	40% Ethanol	1.76	[51]
	Dry fruit	80	40	1:9	Ultrasonic power 250 W	2.86	[52]
	Dry leaves and stem		104	1:38	80% Ethanol	4.01	[53]
	Dry fruit pomace	50.65	29.2	1:34.75		2.52	[47]
Microwave extraction	Dry fruit		25	1:35	Microwave output power 650 W 50% Ethanol	1.72	[46]

**Table 3 foods-12-01373-t003:** Extraction methods of polysaccharides from *S. grosvenorii*.

Method	Materials	Temperature (°C)	Time (min)	Material-Liquid Ratio (g/mL)	Other Conditions	Yield (%)	Ref.
Hot water extraction	Dry fruit	70	150	1:35		6.56	[68]
	Dry fruit	70	180	1:35		14.55	[71]
Enzymatic extraction	Dry fruit	50	55	1:50	Enzyme activity 500 U/g pH 6.0	6.82	[70]
	Dry fruit	50	62		Cellulase dosage 8% pH 5.9	6.64	[72]
Ultrasonic extraction	Dry fruit	70	30	1:30	Ultrasonic power 300 W	6.39	[69]

## Data Availability

The date presented in this paper are available upon request from the corresponding author.

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
