# Peer review of "Siraitia grosvenorii* (Swingle) C. Jeffrey: Research Progress of Its Active Components, Pharmacological Effects, and Extraction Methods"

_foods, 2023, doi:10.3390/foods12071373_

Round 1

Reviewer 1 Report

The manuscript you sent me for review is well-planned and original.

1. There are some grammatical and word spelling errors. Please review the article for language.

2. In the summary and table titles, including the title of the manuscript, only the plant species is written. Is the fruit, leaf or root of this species used in the research, or are the ingredients in the fresh or dried form of these raw materials? Since this information is given on a species basis (especially in tables), it is not understandable. It should be reviewed and corrected.

3. In addition, only the extract yield is available with the extraction methods in the tables. However, in addition to the extract yield, we would like to see in the tables and discussion, in which extraction method, which extraction parameters and in what proportions, and how many of these functional active substances are obtained according to the method. Please add the sections I mentioned above to both the tables and the discussion section.

Best regards

Reviewer 2 Report

The manuscript “Siraitia grosvenorii (Swingle) C. Jeffrey: research progress of active components, pharmacological effects and extraction methods” is a review on the bioactive components, pharmacological effects, and extraction methods of S. grosvenorii, and proposes future development directions. The idea behind the work is interesting; however, a larger study of the previous literature has to be performed, as well as a more critical discussion of the findings has to be developed.

Detailed comments:

- Abstract. Add the main conclusions discussed in the work.

- Extraction methods. In this section, supercritical fluid extraction should be added and described to give an overall idea of the main extraction methods applied to this vegetable matter. In order to introduce this innovative extraction method, see for instance these works: Baldino et al., Journal of Supercritical Fluids, 2018, 131, pp. 82–86; Casas-Cardoso et al., Foods, 2021, 10, 2471; etc…

- Improve English and correct typos.

Reviewer 3 Report

Dear Researchers,

Very nice effort for this compilation. i appreciate your work in alongwith the chosen concept. little modifications and suggestions are incorporated in the attached PDF draft. Moreover, if spare some time, than better to cut short the 4 line long statements, encountered within this review.

Regards;

Round 2

Reviewer 2 Report

The manuscript can be accepted in the present form.